# Automated Mobile Hot Mist Generator: A Quest for Effectiveness in Fruit Horticulture

**DOI:** 10.3390/s22093164

**Published:** 2022-04-20

**Authors:** Dmitriy Khort, Alexey Kutyrev, Nikolay Kiktev, Taras Hutsol, Szymon Glowacki, Maciej Kuboń, Tomasz Nurek, Anatolii Rud, Zofia Gródek-Szostak

**Affiliations:** 1Department of Technologies and Machines for Horticulture, Viticulture and Nursery, Federal Scientific Agroengineering Center VIM, 1-st Institutsky Proezd, 5, 109428 Moscow, Russia; dmitriyhort@mail.ru (D.K.); alexeykutyrev@gmail.com (A.K.); 2Department of Intelligent Technologies, Taras Shevchenko National University of Kyiv, Volodymyrs’ka Str., 64/13, 01601 Kyiv, Ukraine; 3Department of Automation and Robotic Systems, National University of Life and Environmental Sciences of Ukraine, Heroiv Oborony Str., 03041 Kyiv, Ukraine; 4Department of Mechanics and Agroecosystems Engineering, Polissia National University, Stary Boulevard, 7, 10008 Zhytomyr, Ukraine; 5Department of Machine Use in Agriculture, Dmytro Motornyi Tavria State Agrotechnological University, B. Khmelnytsky Ave., 18, 72312 Melitopol, Ukraine; 6Department of Fundamentals of Engineering and Power Engineering, Institute of Mechanical Engineering, Warsaw University of Life Sciences (SGGW), 02-787 Warsaw, Poland; 7Department of Production Engineering, Logistics and Applied Computer Science, Faculty of Production and Power Engineering, University of Agriculture in Krakow, Balicka 116B, 30-149 Krakow, Poland; maciej.kubon@urk.edu.pl; 8Eastern European State College of Higher Education in Przemysl, Ksiazat Lubomirskich 6, 37-700 Przemysl, Poland; 9Department of Biosystem Engineering, Institute of Mechanical Engineering, Warsaw University of Life Sciences (SGGW), 02-787 Warsaw, Poland; tomasz_nurek@sggw.pl; 10Faculty of Engineering and Technology, Higher Educational Institution “Podillia State University”, 32316 Kamianets-Podilskyi, Ukraine; anatoliyrudj@gmail.com; 11Department of Economics and Enterprise Organization, Cracow University of Economics, 31-510 Krakow, Poland; grodekz@uek.krakow.pl

**Keywords:** automated platform, hot mist generator, plant protection products, sprayer, dispersion, operating modes, water-sensitive paper

## Abstract

The study relates to the use of automated plant protection systems in agriculture. The article presents a proprietary automated mobile platform with an aerosol generator of hot mist. Furthermore, the cause of the loss of a chemical preparation in the spraying of plant protection products on the tree crown was determined in the course of field research. A statistical analysis of the results of experiment was carried out and the effect of droplet size on leaf coating density was determined. The manuscript presents a diagram of the degree of penetration of the working solution as it drops into the crown of the tree, as well as a cross-sectional graph of the permeability of the spray from the projection of the fruit tree crown. The most effective modes of operation of the automated mobile platform for spraying plant protection products with a mist generator aggregate were established. Analysis of the results shows that the device meets the spraying requirements of the procedure for spraying plant protection products. The novelty of this research lies in the optimal modes identified by movement of the developed automated mobile platform and the parameters of plant treatment with protective equipment when using a hot mist generator. The following mode parameters were established: the speed of the automated platform was 3.4 km/h, the distance to the crown of the tree was 1.34 m, and the flow rate of the working fluid was 44.1 L/h. Average fuel consumption was 2.5 L/h. Effective aerosol penetration reduced the amount of working fluid used by up to 50 times.

## 1. Introduction

Currently, the use of plant protection products and modern methods of controlling pathogenic bacteria, viruses, fungi, and other organisms that are carriers or pathogens of diseases in agricultural crops, is necessary. The task is urgent due to the great need in the world for high-quality food and an increase in the population of the planet. Biologically and socially mandatory daily minimum consumption of fruits is at least 0.25–0.3 kg, or 90–110 kg during the year, and to ensure a psychologically stable rhythm of human life, it is necessary to almost double this [1]. Global losses of horticultural products from harmful organisms are approximately 30–40% per year, including pests up to 15.8%, diseases up to 10.2%, and weeds [2].

To date, horticultural spraying equipment has been used effectively to mitigate pest infestation. At the same time, the effectiveness of technological measures directly depends on the correct selection of the preparation, the type of treatment and the type of devices for spraying working fluids [3,4]. With this knowledge, it is possible to achieve the most effective results for the procedure [5], and to spray the exact amount of liquid in the right place and the specified time.

For effective and safe use of plant protection products (PPP), it is necessary that the machines and equipment for their application use a high-quality technological process [6,7]. The nature of the distribution of the spray nozzle on the treated surface, its ability to penetrate, cover, and for the PPP to stay on the surface is crucial for effective processing [8]. This largely depends on the sprayers used, on the knowledge and skills to adapt the treatment to specific conditions, and on the accuracy of setting and adjusting the spraying technique [9]. Improper adjustment of machinery and equipment for use in PPP leads to preparation losses, environmental pollution, exceeding the permissible application norms, and a decrease in yield. When planning and carrying out protective measures, it is necessary to take into account the peculiarities of the type of processing, and the specific horticultural crops. Research has established [10,11,12] that best results are based on the retention of droplets on the leaves and the density (coverage area) of the leaves with drops of various sizes. Solving the problem of effective protection of plants from pests and diseases by creating an automated mobile platform with an aerosol generator of hot mist will increase the yield and quality of fruits by reducing the incidence of plants.

The results of the research revealed modes of operation of the robotic platform with a hot fog generator, which made it possible to improve the quality of the technological operation for plant protection products in an autonomous control mode.

## 2. Literature Review and Problem Statement

The literature review is divided into two parts:Study of the influence of plants when spraying with plant protection products, as well as analysis of spraying devices.Research of robotic and automated technologies in horticulture, mathematical methods, hardware and software.

### 2.1. Research on Plants and Effects of Spraying and Spraying Device Analysis

Many scientists from western Europe, America, Russia, India, and other countries have been involved in the investigation of plant protection systems. J. Zabkiewicz, M. Haslett et al. from Rotorua (New Zealand) [13] studied the physical features of applying a plant protection preparation to leaves. In particular, they showed that the causes of drug loss of a preparation carried by air flow to a tree can be related to six factors.

-Vibration of the object by adverse forces such as natural air flows (mechanical action of a branch or trunk of a tree).-Structure of the leaf and the contact area of the droplet with the leaves.-Inclination of the leaves and the position of the leaves relative to the vertical.-Elasticity of the sheet, which affects the change in the position of the sheet from the impact of a concentrated load.-Degree of heating of the working fluid during the procedure.-Sum of different combinations of chemical losses.

To solve this problem a team of researchers from China and Germany (X. Cai, M. Walgenbach, M. Doerpmond et al. [14]) presented a diagram of an installation used to control the injection rate of pesticides using a closed-cycle method. For this, temporary flow meters and an injection speed controller were used. Researchers of the Institute for Applied Plant Protection (H. Nordmeyer, M. Krebs et al., Brown-Schweig, Germany [15]) developed an efficient direct injection system that saves herbicides and reduces environmental impact.

A similar research direction is described in the works of scientists of the Ryazan State Agroecological University, Yu. Kostenko, O. Teterin and others [16,17,18]. In particular, the thermal balance of the hot mist was simulated [16], including the operation of the hot mist generator during stem processing. Models of the distribution of NaCl salt crystals on pallet samplers and a graph of the dependence of the number of salt crystals on the processing speed were drawn and tested in a laboratory installation [17]. The article [18] investigates the temperature scope in the mixing chamber during aerosol seed treatment. The methods of PPP loss from the leaf surface are shown in Figure 1.

The disadvantage of high-volume spraying is low labor productivity. It is closely related to the significant consumption of liquid per unit area. Excess solution flows down in the form of large drops from the crowns of trees to the ground, and part remains on the leaves and fruits in the form of drops and drips on the edges of the leaves, leaving a noticeable trace after drying) Moreover, water and the working fluid must be refilled often.

At the same time, when applying PPP in fungicidal and insecticidal treatments, especially contact and desiccation preparations, it is better to use fine dispersion sprayers with a particle diameter not exceeding 30 microns, e.g., a mist generator. This is to ensure good penetration and better deposition of the solution with the maximum coverage area of the plant organs in tiers (Figure 2).

Cold and hot mists differ with the size of the droplets:-Conventional spraying (through nozzles)—100–300 microns.-Large-volume spraying—300 microns.-Low-volume spraying—100–200 microns.-Ultra-high volume spraying—100 microns.-Cold mist—50–100 microns.-Hot mist—1–20 microns.-Dry mist—1–5 microns.

The main difference between cold and hot types of mist is the size of the resulting particles. Small droplets easily penetrate the crown of trees, evenly distributing throughout the treated surface. However, due to heating followed by evaporation, this type of generator is not compatible with all preparations. The consumption of the working fluid during aerosol treatment is ten times lower than during spraying. In addition, labor productivity increases significantly, and the quality of work improves [16].

The following is an analysis of existing machines and units for spraying plants, which allows selection of the required parameters for the plant under study. The analysis showed that professional mechanized outdoor application of plant protection products can be carried out at speeds of 6–8 to 14–16 km/h, or more. The consumption rate of the working fluid is 100–150 L/ha for various types of sprayer, depending on the target object, the state of culture, the weather, and other conditions. To date, in Russia, slot sprayers with a flat nozzle of one or two sizes are used to spray horticultural crops, with a large variation in the size of the drops produced. The speed of movement during the spraying operation averages 8–15 km/h with a working fluid consumption of 200–300 L/ha or more. In mobile sprayers it is reduced to 50–80 L/ha; in this case the quality of treatment can be questioned, especially in adverse weather conditions, when incorrectly selected sprayers, in terms of their expiration dates, wear, and other factors. Rotating disk nozzles have been most broadly used in Russia in ONM-600 and SUMO-24 rod sprayers. Such nozzles are installed horizontally on rod sprayers and the droplets sprayed are deposited gravitationally. This, however, leads to a decrease in spraying efficiency and an increase in the drift of droplets of the working fluid [16]. To reduce the drift and improve the quality of spraying, forced deposition of preparations is used, for which rotating nozzles are installed on rod sprayers, together with low-power fans. 

To increase the efficiency and cost effectiveness of protective treatments (to reduce the consumption of PPP), only high-tech equipment should be used with pesticides having low ecotoxicity, low consumption rates, and formulas that do not cause sprayer clogging. Taking the above into account, the design features of modern rod sprayers and methods to increase their productivity are analyzed, and the main directions for improving the designs of nozzles and sprayers in general are identified in the paper. The currently used technical means and spraying devices to protect fruit plantations from diseases and pests in industrial horticulture have a number of significant disadvantages, such as low productivity, high costs of working solutions, fuels and lubricants, high time and labor costs, and poor processing quality. The relationship between the structure of the fruit tree crop and the performance of an axial fan sprayer in orchards is described in an article by UK researchers P.J. Walklate, G.M. Richardson [19].

Analysis of existing technical methods and devices for applying plant protection products has revealed the following disadvantages.

-High consumption of pesticides.-Uneven distribution of pesticides on the surface of plants.-Low penetration ability into the crown of trees.-Low productivity and capture width, and the inability to control the process when speed and direction of wind changes and updrafts appear.-The need to introduce protective sanitary zones around the treated areas, as well as jeopardy to beneficial insects and birds.

The joint work of Belgian, Australian and New Zealand researchers, M. Massinon, N. De Cock et al., is also of interest [20]. They describe the effects of spraying three formulations on four plant species: beans, avocado, cattle grass, and cabbage. A crawler sprayer was used equipped with a high-speed camera. To measure the size and velocity of drops, measurements were carried out by image processing based on an algorithm to test several hypotheses. Belgian researchers, D. Dekeyser, T. Ashenafi, A. Duga [21,22], modeled and developed optimal modes of operation of horticulture sprayers. A computer simulation of hydrodynamics was also carried out, as well as a 3D model of this device and its application in horticulture sprayers with a pneumatic drive. Polish researchers, G. Doruchowski, P. Marucco et al. [23], developed an application called an Environmentally Optimized Sprayer (EOS) to support sprayer manufacturers and pesticide application specialists in the selection and operation of sprayers. The application evaluates the risk reduction potential of sprayers based on their technological performance in five risk areas representing pollution sources: internal and external pollution, filling, spray loss and drift, and leftovers. 

In addition to herbicide spraying, the literature has established the effect of ozonation on the health benefits and antioxidant properties of Honeoye strawberries, described in the work of Polish researchers [24,25]. The evaluation of plant quality using the example of wild and cultivated green tea from different regions of China is described in an article by Polish scientists [26]. These studies were used in the development of this project.

### 2.2. Research in the Field of Automation and Robotization Technologies in Horticulture

This section provides an overview of hardware and software solutions for automating and robotic gardening, features of which can be used to create a robotic platform for applying hot fog in an apple orchard.

An automated electric drive for a sprayer tested on citrus fruits is described in the articles of American researchers L. Khot, R. Ehsani, N. Pai et al. [27,28]. Regulation of the spraying speed depending on various meteorological conditions is described, as well as effects related to the density of the foliage of the trees.

An article by Indian researchers [29] describes an automated plant for the treatment of plant leaves with pesticides to prevent diseases. The robot is propelled with a L293d engine driver, a processor and an integrated Raspberry Pi3 system. Python code was developed for machine learning training of the robot based on predefined images. The robot can be controlled remotely to protect the farmer from exposure to pesticides.

Researchers of plant diseases at the National University of Life and Environmental Sciences of Ukraine, T. Lendiel, I. Bolbot et al. proposed a method [30] for contactless visual assessment of plants in a greenhouse. The study involves photographing plants with a specialized unit and recognizing them using wavelet analysis. This technology, was used as a means of contactless collection of information, allows assessment of the growth and condition of plants in a greenhouse, and predicting their development using mathematical transformations. When recognizing photo images of plants, the condition of a particular plant can be assessed, including disease detection, in which case the staff is informed. The software and hardware of the phytomonitoring unit in the greenhouse is based on the LabVIEW software environment and Arduino hardware. This subsystem was also tested at the JSC Greenhouse Combine production plant in the village of Kalinovka, Kiev Region (Ukraine) [31]. 

Next, we consider the existing developments of robotic platforms for working in gardens. Smart automated systems for plant protection product control in orchards, based on a fuzzy algorithm, were described in an article by Slovenian researchers P. Berk, A. Belšak et al. [32]. A decision-making model most suitable for real-time PPP distribution was obtained. The advantage of this development is the use of a mobile computer that implements a solution algorithm using information obtained using ultrasonic sensors. A research group from Italy developed a solution based on a reconfigurable vehicle with a high degree of automation for the distribution of plant protection products in vineyards and greenhouses [33]. A robot sprayer based on a car equipped with a “smart spraying system” was developed to optimize spraying operations. The human-machine interface allows operators to schedule spraying treatment, monitor the operation from a safe distance, and create a final report. The architecture consists of four main subsystems: a mobile crawler robot, a location and navigation system, a smart spray system, and a human-machine interface.

Researchers from Russia and Ukraine, with the participation of the authors of this article, D. Khort, A. Kutyrev, N. Kiktev, have developed several automated platforms for various purposes for use in agriculture [34,35,36,37,38]. One of them is an automated device for hydraulic weed removal [35], and another is a control system for an automated vehicle [37]. An automated platform for strawberry picking is described in [36], a digital vision system for strawberry recognition in [35] and for apple recognition in [38]. An automated device for the differential application of herbicides in the trunk zone of fruits, similar to the topic of this manuscript, is described in [34]. However, the above work was for weed crossing, and this article aims to find the optimal parameters for a stand-alone hot mist aerosol platform, which has its own characteristics. Interesting insights into the modeling and design of berry and fruit picking robots using the Active-HDL software environment were described in a work coauthored by one of the authors of this article and by a team of researchers from Taras Shevchenko National University of Kyiv (Ukraine) [39]. A hardware and software complex for managing an agricultural object using Arduino and LabView software is presented in [40].

A mobile automated platform developed at the National University of Life and Environmental Sciences of Ukraine [41] capable of moving around the greenhouse using technological guidelines, provides monitoring of the main parameters of the atmosphere in the greenhouse, as well as phytomonitoring, including product quality, detecting diseases, minimizing the route inside the greenhouse, avoiding obstacles, and ultimately contributing to the economic profit of the enterprise.

Of interest is an automated platform for testing the tightness of large-diameter pipelines developed by Spanish researchers A. Ibarguren, J. Molina et al. [42]. Robot path planning is done in two steps: 1—a general plan is created in a topological graph using Dijkstra’s algorithm, which is the basis for low-level planning; 2—the robot moves locally using local metric maps and a search-based planning algorithm. Planning is done in the dimensions x, y, and θ, resulting in smooth trajectories based on the orientation of the robot. Using the information provided by the GPS/IMU sensor, the robot executes the planned path, and the system uses the local detector to avoid obstacles and to get as close as possible to the initial path. 

In research by Japanese scientists S. Hata, T. Hiroyasu, J. Hayashi [43], a vision system for a robot was presented used in the process of transplanting plants and suitable for small seedlings with an unstable shape and size. Complete morphology of plants for biohybrids of robot plants was described in the studies of scientists from Germany, M. Wahby et al. [44]. 

Australian researchers P.J Sammons et al. developed a robot sprayer for greenhouses [45]. It uses inductive infrared proximity sensors. This ensures safe and trouble-free operation of the robot inside the greenhouse. The advantage of the development is the use of a microprocessor web host to control the movement of the robot from a remote location. Norwegian researchers L. Grimstad, R. Zakaria et al. [46] presented an autonomous robot for operation in greenhouses that automatically estimates the location of rails on the map and locates them using its onboard sensors. The use of sprayers in agriculture was reported by Indian researchers in the 1990s [47]. Some reports describe hi-tech methods of plant protection [48], especially artificial intelligence-based, automated platforms, as well as sensors and microcontrollers used for irrigation, watering, plant protection and other agricultural treatments [48].

Researchers from the National University of Life and Environmental Sciences of Ukraine, N. Pasichnyk, O. Opryshko, D. Komarchuk et al. [49,50,51,52], investigated plant diseases and predicted leaf damage at an early stage by recognizing images from unmanned automated vehicles (UAVs). The authors used an algorithm developed in Python to identify the path of the vehicle, the contours of areas with similar values of vegetation index, and main signs of plant stress. The authors also investigated nitrogen nutrition in wheat plants. A specialized neural network and spectral indexes were used as mathematical tools. 

Ukrainian researchers S. Yermakov, T. Hutsol et al. analyzed technical solutions for planting using energy willow. The results can be useful in the design of automated platforms for plant protection treatments [53]. Italian researchers [54] describe an identification system based on ultrasonic sensors for use when spraying pesticides in apple orchards and vineyards. Improving the energy efficiency of the spraying system by creating a sprayer with automatic remote control of air flow was investigated in an article by Polish researchers R. Hołownicki, G. Doruchowski et al. [55,56]. The system consists of three subsystems.

A crop health sensor, which determines the condition of fruit crops.A crop identification system, which determines the size and density of the tree crown.An Environmentally Dependent Application System (EDAS), which determines the environmental conditions during spraying.

The EDAS system automatically adjusts spray parameters such as droplet size and airflow rate based on wind speed and direction, and the position of the sprayer in relation to sensitive areas such as neighboring orchards, surface water, and drinking wells, to protect these areas from pollution.

Assessment and forecast of the risk of investments in investment projects, including those related to plant protection, are described by Ukrainian and Polish researchers A. Tryhuba, T. Hutsol, M. Kuboń, et al. [57,58]. 

Well-known solutions regarding robotic hardware and software technologies for plant protection and other garden work are useful for the implementation of the robotic platform described in this article and for the further development of the project.

### 2.3. Purpose and Objectives of the Study

The purpose of the research was to assess the quality of spraying plant protection products on apple trees, as exemplified by an aerosol generator of hot mist installed on an automated mobile platform, and to determine the effective operating mode of the platform.

The research task was to develop and implement a highly efficient automated horticulture sprayer capable of ultra-small-volume spraying of fruit trees with a heated working fluid using a mist generator. Relevance is based on the great global demand for high-quality food products caused by the global population growth. According to the UN Department of Economic and Social Affairs, the world population will increase to 8,322,701,000 people by 2030. Solving the problem of effective plant protection from pests and diseases will increase the yield and quality of fruits in the orchard.

## 3. Materials and Methods

A universal automated mobile platform (URP) developed at the Federal Scientific Agroengineering Center VIM (Moscow, Russia) with an aerosol mist generator was used for the research (Figure 3).

The technical characteristics of the automated mobile platform (URP) are presented in Table 1.

To apply plant protection products, a sprayer was used to distribute the heated liquid at a particle diameter corresponding to mist (mist generator). Artificial mist is formed as a result of heating (by the combustion of gasoline) of toxic chemicals in the form of a working fluid and then evaporation and condensation under cold air. The liquid is then fed to the nozzle under pressure and sprayed into small particles, not exceeding 30 microns. The technical specifications of the aerosol mist generator are given in Table 2.

To determine the most effective modes of operation of an automated platform in an industrial plantation of a 5-year-old apple orchard, a factor experiment was conducted in the scientific and production testing department of the Federal Center for Horticultural Research for Breeding, Agrotechnology and Nursery.

As a response function during the experiment, the coverage density of leaves (registration cards) with the working fluid (%) was assessed. When applying PPP, the quality of the operation is affected by the following factors: speed, km/h, distance to the tree crown, m, working fluid consumption, L/h, fuel consumption (L), total amount of water (L), total amount of PPPs used (L), and share of plant protection products dispersed in the environment (%). Next, we identify significant independent factors that included in a real experiment.

The total amount of PPP used was a constant dose of plant protection products, as required by agrotechnological requirements. Fuel consumption does not affect the quality of the technological operation, since the purpose of the research was to determine the most effective operating modes of the platform for the high-quality application of plant protection products. The amount of water is not an independent factor, as it is related to the flow rate of the working fluid. The share of plant protection products dispersed in the environment (%) is taken as a constant value, depends on the chemical composition of the agent used, on the droplet size, and on climatic conditions (wind speed during the experiment ranged from 0.5 to 1 m/s). Thus, by sifting out insignificant and dependent factors, we determined the factors included in the real experiment: the speed of movement (*X*_1_), km/h, the distance to the tree crown (*X*_2_), m, the flow rate of the working fluid (*X*_3_), L/h.

The interval of variation of factors was chosen based on preliminary field studies. The experiment was carried out twice [59]. The factor plan of the experiment with operating modes is shown in Table 3.

The output parameter is the density of the leaf coating with the working fluid (%). Since hot mist droplets have a higher temperature than the objects being treated, they evaporate quickly. As a working solution, the fungicide “Horus” was used, at 2 g/10 L of water. Syngenta water-sensitive paper (made in Switzerland) was used to analyze the coating density and count the number of drops per square centimeter. Water-sensitive paper was attached to apple leaves at heights of 1 m, 1.5 m, and 2 m and at different depths with a step of 0.3 m in the crown of the tree, shortly before spraying. Water sensitive paper was used as a registration card to determine the density of PPP leaf coverage. After spraying, the registration cards were colored dark blue as a result of drops of PPP working fluid on their surface (Figure 4). This method has been confirmed by Chinese colleagues and described in [60].

Statistical processing of the results of the field experiment was carried out, and mathematical modeling was carried out using the method of planning field experiments in the PlanExp v.1.0 software. The plan of the factor experiment, with the values of input and output factors in coded (columns 2–4) and natural (columns 5–7) form is given in Table 4.

Linear interpolation was determined, the adequacy of the model was checked according to the Student and Fisher criterion, and the variance of reproducibility in parallel experiments was found.

The efficiency of the procedure of PPP application is significantly affected by weather conditions, the deviation of which can lead to a multiple decrease or lack of efficiency and can also harm the environment. During the field experiment, the air temperature varied from 18 to 21 °C, the humidity ranged from 75 to 80%, the wind speed did not exceed 4.5 m/s, and precipitation was absent, including mist and dew, which correspond to the agrotechnical requirements of spraying orchard plantings.

## 4. Results

As a result of the field experiment, the density of the droplet coating was determined when introducing PPP in various operating modes of the mist generator (Figure 5). In the experiment, one tree was used and the number of card entries was 39 pieces. This method is used to determine the quality of spraying, as described in [60].

It was revealed that the uniformity of the distribution of drops in height, length and depth of penetration of the treated area of the tree corresponded to the distribution norm according to agrotechnical requirements [61].

The uniformity of the distribution of hot mist droplets along the height of the tree crown was found to be mainly related to the angle of inclination of the generator nozzles relative to the horizontal plane. Moreover, the uniformity of the distribution along the length of the crown depended mainly on the speed of the air flow generated by the hot mist generator and the linear size of the droplets.

To analyze the results obtained, water-sensitive paper analysis was performed using a magnifying biconvex glass. Calculation of the average number and diameter of drops per cm^2^ of each card was carried out (Figure 6).

Analysis of the results allowed determination of the tiered degree of penetration of drops of the working solution inside the crown of the tree (Figure 7). The results are presented in the form of a diagram of the quantitative assessment of the penetration of drops inside the tree crown when PPP are applied on one side of the row. On the left is a frontal view (A) of a tree, on which registration cards are placed-in three zones (outer 1, middle 2 and inner 3, as well as along the trunk). Each zone is located in three tiers (upper 1–3, middle 1′–3′, bottom 1′′–3′′) and on the ground. A total of 39 registration cards were used in the experiment (for one repetition). The solution was applied to the tree from the left in the amount of Q (L/ha). On the right is a top view (A-A) of the droplet penetration quantification chart. The measurement scale is the number of drops (0–700). The lines of the number of drops of the solution in the tree zones (1, 2, 3) on each of the three tiers are marked with colors. As can be seen from Figure 7, the largest number of droplets of the solution falls from the side of the aerosol hot fog generator.

This diagram corresponds to the appearance of the registration cards (Figure 5). Card 13 has the most drops (the most pronounced blue color). The color of these registration cards corresponds to the diagram. Cards 13–16 represent the middle tier in the outer zone; cards 17–20 represent the middle tier in the middle zone; cards 21–24 represent the middle tier in the lower zone.

The number of drops on the leaves in different zones of the tree was determined. A graph was built to estimate the number of drops that penetrated the crown of the tree during a single pass of the robotic platform and the introduction of plant protection products on one side of the row.

It was found that the number of drops per cm^2^ of registration cards on the middle tier of the tree was 18.1% more (684) than on the upper tier (579) and 21.7% more than on the lower tier (562). In the registration cards located on the ground (37–39), the number of drops per cm^2^ was 31.3% lower compared to the average tier (521).

A graph of the permeability of hot fog droplets was plotted according to the projection of the tree crown in the transverse plane (Figure 8). On the right is a robotic platform that moved around a tree at a speed of V, km/h. In the upper part of the platform at an angle α (assumed to be constant) there is an aerosol hot fog generator that sprays PPP at a certain amount Q (L/ha) located at a distance H to the axis of the tree 0-0.

Places for hanging registration cards ae located on planes of equal penetration depth of the spray 1-1, 2-2, 3-3 from the side of the aerosol fog generator and 1′-1′, 2′-2′, 3′-3′ from the back side. On the lower horizontal axis the distance from the center (m) was Δ-deviation from the axis of the row.

The vertical axis is the droplet density, pcs/cm^2^. The experimentally obtained graph (blue line) shows the density of drops (the number of drops per cm^2^) at a distance from the tree trunk 0 m (row axis) to ±2 m from it (tree crown). It can be seen from the graph that the density of drops was the highest in the upper tier of the crown from the side where the robotic platform was located (in the zone of dispersion of plant protection products), and the smallest was in the lower tier on the back side of the row.

The density of drops at the point of intersection of the curve with the *y*-axis was established. For one pass of the robotic platform along the aisle in a plane of equal penetration depth passing through the axis of the row, the droplet density was 480 pcs/cm^2^. When processing this row on the other hand, the number of drops doubled.

The density of droplets at the point of intersection of the curve with the ordinate axis was established. In one pass of the automated platform along the aisle in a plane of equal penetration depth through the axis of the row, the density of droplets was 480 pcs/cm^2^. When processing this row, on the other hand, the number of drops doubled. The results of statistical processing of the factor experiment data are shown in Table 5.

The coefficients of the mathematical model were found. An equation for the mathematical model was obtained:y = (649.143) + (−58.12) · *X*_1_ + (−59.072) · *X*_2_ + (2.516) · *X*_3_ + (−49.1) · *X*_1_^2^ + (−68.683) ·
*X*_2_^2^ + (−8.431) · *X*_3_^2^ + (−5.235) · *X*_1_ · *X*_2_ + (1.107) · *X*_1_ · *X*_3_ + (10.877) · *X*_2_ · *X*_3_

Analysis of the mathematical model according to the Fisher criterion allowed us to determine its accuracy (F = 2.77 < Table = 3.48).

As a result of the transformation, three variants of the mathematical model were obtained for y = f (*X*_2_, *X*_3_) for *X*_1_ = const, y = f (*X*_1_, *X*_3_) for *X*_2_ = const and y = f (*X*_1_, *X*_2_) for *X*_3_ = const. A mathematical model taking into account a constant factor takes the form:

At *X*_1_ = const:y = (649.143) + (0) + (−59.072) · *X*_2_ + (2.516) · *X*_3_ + (0) + (−68.683) · *X*_2_^2^ + (−8.431) ·
*X*_3_^2^ + (0) · *X*_2_ + (0) · *X*_3_ + (10.877) · *X*_2_ · *X*_3_

At *X*_2_ = const:y = (649.143) + (−58.12) · *X*_1_ + (0) + (2.516) · *X*_3_ + (−49.1) · *X*_1_ ^2^ + (0) + (−8.431) ·
*X*_3_ ^2^ + (0) · *X*_1_ + (1.107) · *X*_1_ · *X*_3_ + (0) · *X*_3_

At *X*_3_ = const:y = (649.143) + (−58.12) · *X*_1_ + (−59.072) · *X*_2_ + (0) + (−49.1) · *X*_1_ ^2^ + (−68.683) ·
*X*_2_^2^ + (0) + (−5.235) · *X*_1_ · *X*_2_ + (0) · *X*_1_ + (0) · *X*_2_

It was established that the extreme of the response function of the mathematical model was within the range of varying factors. A graphical interpretation of the function of three variables and a projection diagram of three-dimensional response surfaces on the plane is shown in Figure 9.

The values of the extremes of the response function and the corresponding values of the factors in encoded and natural form were determined (Table 6).

Statistical analysis of the data of the conducted factor experiment allowed us to determine that the following modes were most effective for performing the operation of applying plant protection products using a mist generator installed on an automated platform: speed of the automated platform = 2.1 km/h; distance to the crown of the tree from the nozzle of the mist generator = 1.28 m; flow rate of the working fluid = 39.75 L/h. The average fuel consumption was 2.5 L/h.

## 5. Discussion

Study of the distribution of hot mist droplets on the crown of a tree showed that the greatest droplet coverage of the treated surface occurred in the immediate vicinity of the hot mist generator, while the average value of the number of drops per 1 cm^2^ of the crown of the tree was 507, with an exposure time of 10 s. The average size of the droplets on the leaves of the tree was 26.6 microns. The coefficient of variation of the droplet size was υ = 32.4%. This indicates great influence of weather conditions on the uniformity of the droplets obtained as a result of changes in air temperature, humidity, and wind speed. It was established that the use of a hot mist generator allows reduction of the amount of PPP used by up to 50 times, which leads to a reduction in time costs. The largest droplet size was observed at the edges of the tree crown, due to the inertial distribution of the hot mist droplets.

As a result of the research, three reasons for the loss of the working solution were determined.

-Large drops fall off the sheet because they are heavy.-Very small droplets evaporate in the air and do not reach the leaf.-Small drops are also blown away by the wind and also do not fall on the leaves.

Preliminary data were obtained in laboratory conditions using an artificial tree in the absence of air currents. Registration cards were placed at a distance of 0.5–2 m from the hot fog nozzle.

Knowing the flow rate of the working fluid, Q, and changing the processing time, t, the number of droplets of various sizes that settled on the registration cards was measured. Then, the results of the study in the field (apple orchard) were compared with the data obtained in the laboratory. The loss of large droplets of plant protection products that drained from the leaves was determined experimentally by measuring the volume of the flowing working fluid.

It was found that up to 8% of the losses of the working fluid was due to droplets smaller than 10 microns that evaporated in the air, up to 12% of the losses were due to droplets carried outside the processing zone by air currents, and up to 14% were losses of large droplets, more than 100 microns, that did not stay on the sheet due to gravity. Each of these indicators was evaluated separately in the field during preliminary studies by measuring the total number of drops released per unit time.

The research allowed us to determine the main advantages and disadvantages of using a hot mist generator installed on an automated platform. The advantages include low consumption of PPP per 1 ha, high uniformity of distribution of hot mist droplets on the surface of plants, the possibility of processing trees at a considerable distance from the nozzle, high productivity and large width of capture, and reduction of processing time. The disadvantages are the inability to control the procedure with a significant variation in wind speed and direction, the presence of ascending air flows, the need to introduce additional protective zones around the treated areas, and incompatibility with all types of preparations due to heating of the working fluid.

## 6. Prospects for Further Research

The authors plan further tests of the automated platform to investigate the possibility of its use not only in orchards, but also in greenhouses and in various climatic zones.

Preliminary analysis showed that effective aerosol penetration allowed using up to 50 times less water, which helps to reduce time expenditures (up to 1 h for processing 5000 m^2^). The peak density of droplets in the air occurred after 35 min of operation in a greenhouse with an area of 1000 m^2^ (Table 7).

The authors are working on improving the robot’s motion algorithm for aerosol spraying, using various methods of leaf and fruit recognition for more accurate and efficient leaf processing. Reducing morbidity and increasing yields in the orchard with the use of the developed device will also be investigated. Work is planned to link the robot’s movement to the geographical coordinates of each tree in the orchard row using GPS navigation.

The prospects of using automated platforms with aerosol sprayers are important in the fight against infections; for example, the use of ultra-small volume insecticidal aerosols sprayed from devices installed on vehicles to create cold mist. They are used to combat viral strains during outbreaks of fever and Zika virus, as described in an article by French researchers [62]. In addition, these developments can be used in emergency or epidemic situations, and for seasonal control of flying insect pests or vectors, where it is necessary to use spatial spraying [63].

Generators that produce cold or hot mist have a great advantage over previously used methods of disease, virus and insect prevention (spraying, watering). However, the effectiveness of these devices is determined by the development and use of new insecticides that are effective against insects and viruses, and safe for humans.

For automatic setting of operating parameters, an automated platform employing LIDAR [64,65] can be used to accurately control the distance between the spray gun and to the tree crown and equip the hot mist generator with an additional drive (linear actuator) to change angle of inclination. This will reduce the consumption of working fluid, improve the quality of the operation and minimize human involvement.

Our results show an increase in the efficiency of the technological operation of spraying plant protection products using a robotic platform with a thermal fog generator, resulting in up to 15% reduction of costs due to saving of plant protection products when compared with current widely used methods [66,67]. This was done by assessing the performance based on optimal work schemes and productivity gains when performing offline operations.

## 7. Conclusions

Studies of plant treatment methods using a hot mist generator with an automated platform for automated operation in an intensive-type orchard were carried out. The results of field research indicate that the device meets the requirements for applying plant protection products with sufficiently high technological indicators.

It was found that up to 8% loss of PPP occurred with droplets of less than 10 microns because these evaporated in fallout. Up to 12% loss on fallout occurred because drops were carried outside the growth zone of the plant, and up to 14% loss occurred with droplets of more than 100 microns because they were not kept on sheet under stress. In addition, studies showed that more than 95% of the droplets in the greenhouse settled within a day. The remaining 5% showed up in the park.

The novelty of this study lies in the identified optimal modes of movement of the developed automated platform and the parameters of plant treatment with a PPP (Horus) when using a hot mist generator.

Unlike other studies that carried out processing in various modes, in this study the most optimal mode was established using the factor analysis method. Effective modes of operation were identified for performing the procedure of applying plant protection products in an automatic mode using an automated platform that has the ability to move in orchards according to a task map.

The following mode parameters were established: speed of the automated platform = 3.4 km/h; distance to the crown of the tree = 1.34 m; flow rate of the working fluid = 44.1 L/h. The average fuel consumption was 2.5 L/h. Effective aerosol penetration reduced the amount of working fluid used by up to 50 times.

The authors confirmed the possibility of performing the procedure of applying plant protection products in a fully autonomous mode, with automatic adjustment of operating parameters.

## Figures and Tables

**Figure 1 sensors-22-03164-f001:**
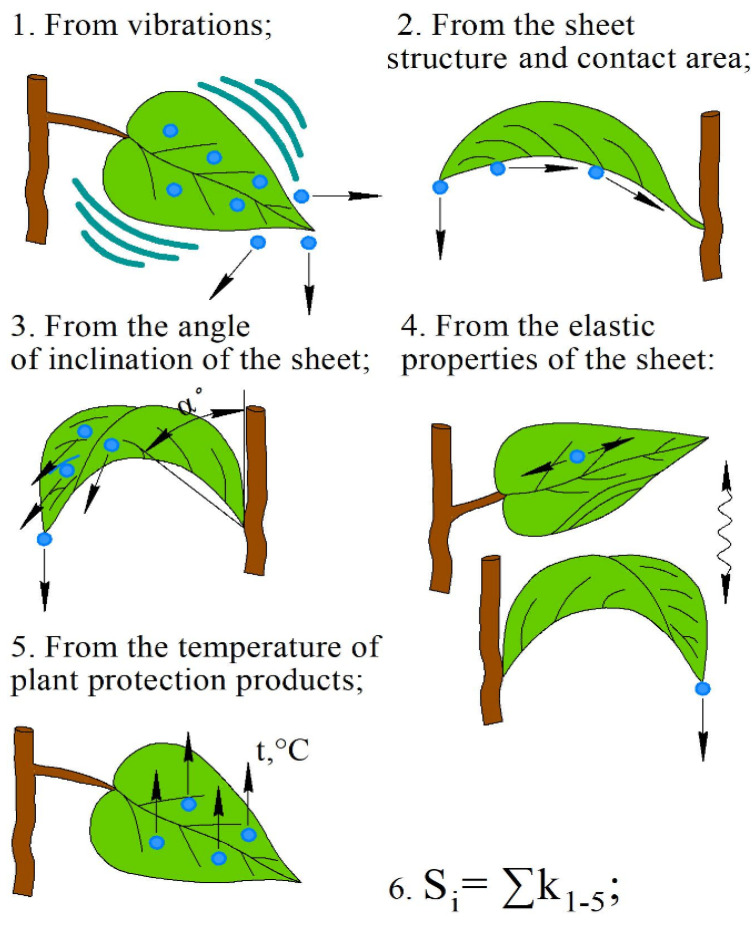
Methods of PPP loss from the leaf surface.

**Figure 2 sensors-22-03164-f002:**
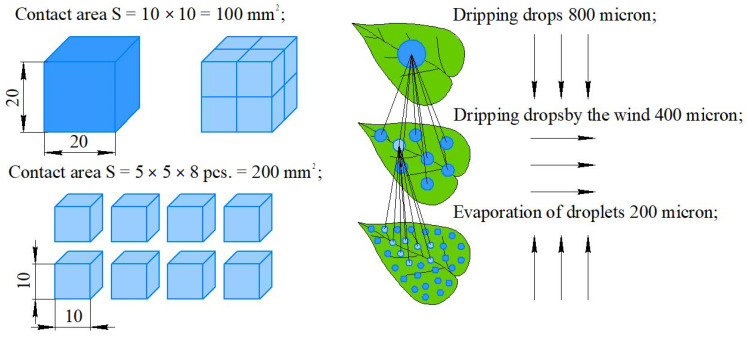
Effect of droplet size on coating density.

**Figure 3 sensors-22-03164-f003:**
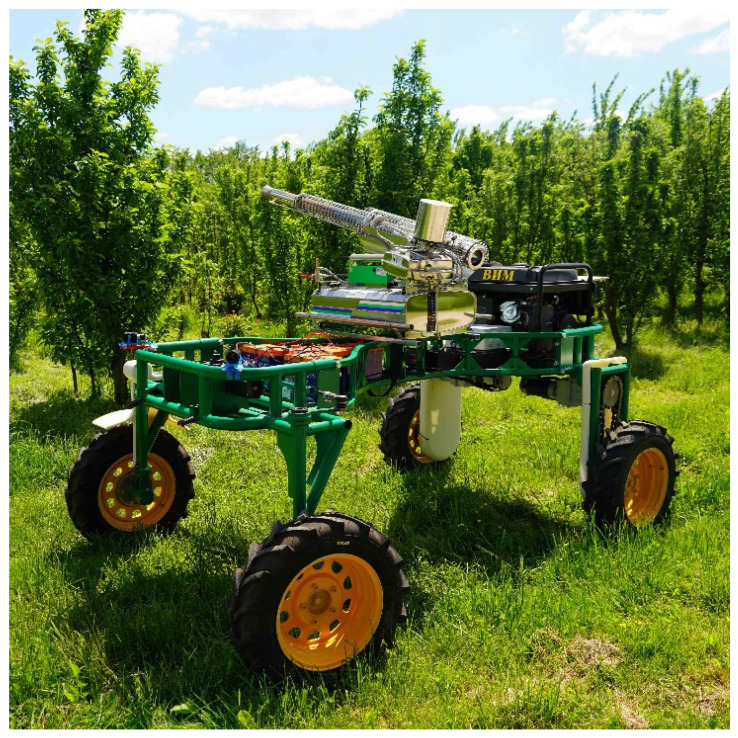
Automated platform with an aerosol mist generator installed.

**Figure 4 sensors-22-03164-f004:**
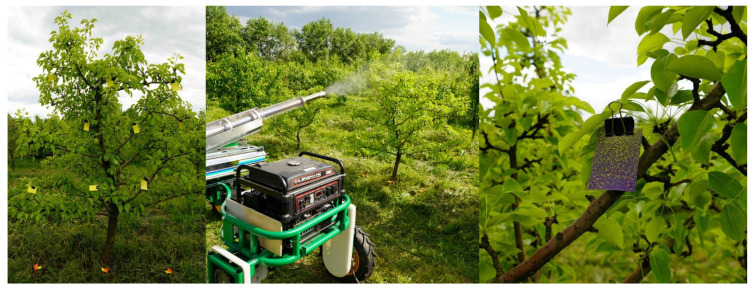
Pieces of water-sensitive paper fixed on a tree after contact with drops of aqueous PPP solution.

**Figure 5 sensors-22-03164-f005:**
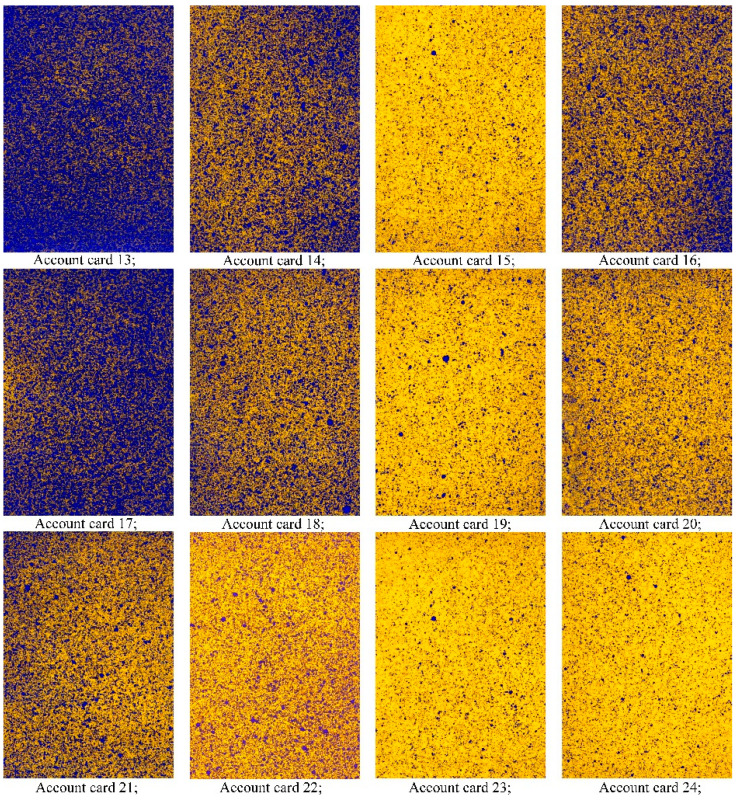
Account registration cards of the middle tier at a platform speed of 1.5 km/h.

**Figure 6 sensors-22-03164-f006:**
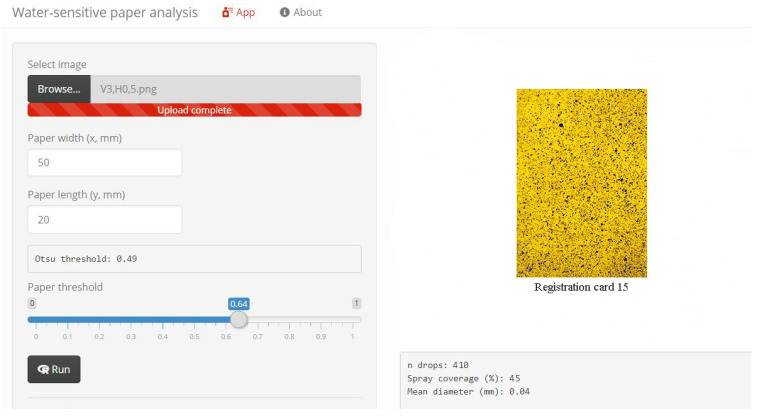
Analysis of the results obtained in the online service. “Water-sensitive paper analysis”.

**Figure 7 sensors-22-03164-f007:**
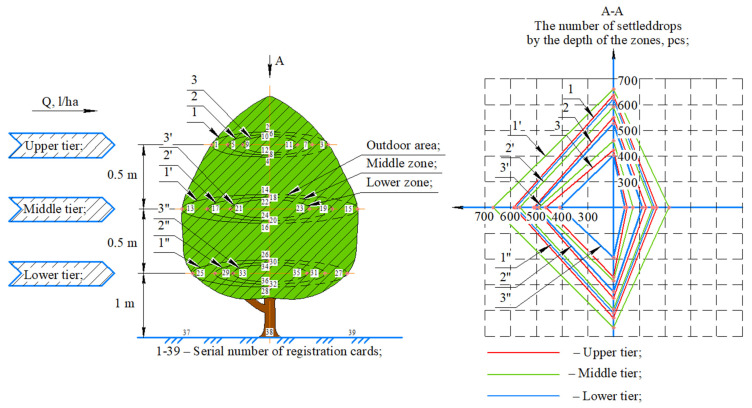
Layout of the registration cards and diagram of the degree of penetration of drops of the working solution.

**Figure 8 sensors-22-03164-f008:**
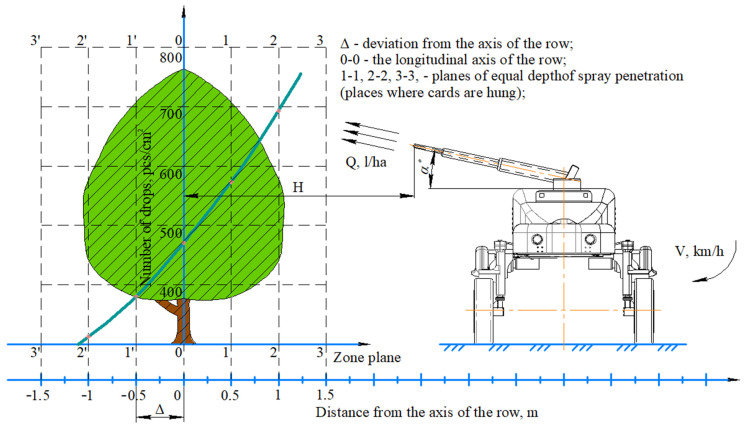
Graph of the permeability of hot mist droplets on the projection of the tree crown.

**Figure 9 sensors-22-03164-f009:**
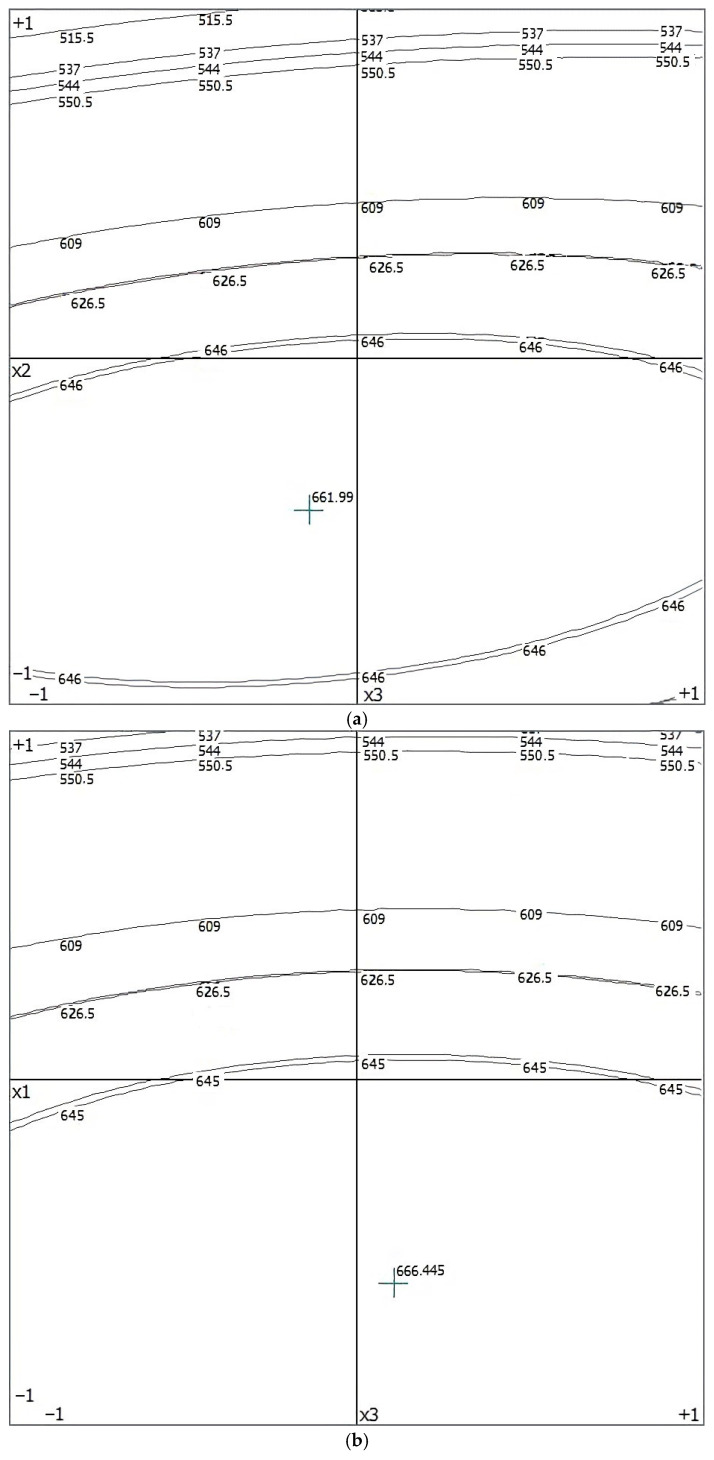
Projection graphs of three-dimensional response surfaces on the plane according to the optimum: (**a**) y = f(*X*_2_, *X*_3_) at *X*_1_ = const, (**b**) y = f(*X*_1_, *X*_3_) at *X*_2_ = const, (**c**) y = f(*X*_1_, *X*_2_) at *X*_3_ = const.

**Table 1 sensors-22-03164-t001:** URP Technical specifications.

Name	Indicator Value
Machine dimensions, not more than, mm	
length	2800.0
width, adjustable	1840.0; 1960.0; 2080.0
height, no more	1600
Track width, base, mm	
front wheels	1800
rear wheels	1800
base	1900
Curb weight, kg	850
Load capacity, kg	500
Ground clearance, mm	1200
The smallest turning radius, not more than, mm	3500
Translational speed, km/h	
working	1–6
transport	8
Surmountable ascent, hail.	
Entrance angle, deg.	20
Exit angle, deg.	18
Power plant	
type	Gasoline generator
generator power, W	5500
power grid voltage, V	48
Drive	
type and quantity	Electric motor with two-stage planetary gearbox, 2 pcs.
power, kW	0.6–0.8 kW
Management system	Remote control with technical vision. The autonomous control system contains an optical range finder (LIDAR), four video cameras, laser distance sensors, GPS navigation station, on-board computer
Brake system	Electric motor braking

**Table 2 sensors-22-03164-t002:** Technical characteristics of the aerosol mist generator.

Name	The Value of the Indicator
Dimensions, cm	198 × 62 × 58
Engine type	gasoline, jet-pulse
Power of the combustion chamber, W	36,775
Gas tank capacity, L	5.5
Fuel consumption, L/h	4
Tank capacity for solution, L	60
Working fluid consumption, L/h	up to 40
The temperature of the mist at the outlet of the nozzle, °C	40–60
Aerosol particle size, microns	1–30
Mist penetration range, m	up to 80

**Table 3 sensors-22-03164-t003:** Conditions for planning a factor experiment.

	Lower Level (−1)	Basic Level (0)	Upper Level (+1)	Variation Interval	Name of the Factor
*X* _1_	1.5	3	4.5	1.5	Driving speed, km/h
*X* _2_	1	1.5	2	0.5	Distance to the tree crown, m
*X* _3_	20	40	60	20	Working fluid consumption, L/h

**Table 4 sensors-22-03164-t004:** Planning a three-factor experiment.

Experience Number	Planning Matrix			Values of Variables			Values of Variables
	*X* _1_	*X* _2_	*X* _3_	Driving Speed, km/h	Distance to the Tree Crown, m	Working Fluid Consumption, L/h	The Density of the Coating of the Leaves with the Working Fluid, %
1	2	3	4	5	6	7	8
1	−1	−1	−1	1.5	1	20	632
2	+1	−1	−1	4.5	1	20	525
3	−1	+1	−1	1.5	2	20	515
4	−1	−1	+1	1.5	1	60	631
5	−1	0.19	0.19	1.5	1.595	43.8	667
6	0.19	−1	0.19	3.2	1	43.8	624
7	0.19	0.19	−1	3.2	1.595	20	624
8	−0.29	+1	+1	2.65	2	60	557
9	+1	−0.29	+1	4.5	1.355	60	556
10	+1	+1	−0.29	4.5	2	34.2	390

**Table 5 sensors-22-03164-t005:** Statistical processing of field experiment data.

Name	Indicator Value
Number of degrees of freedom	10
Student’s Criterion	2.23
Variance of the adequacy of the mathematical model	432.6
Degrees of freedom at significant coefficients	5
Tabular value of the Fisher criterion	3.33
Calculated value of the Fisher criterion	1.43

**Table 6 sensors-22-03164-t006:** The obtained values of factors in coded and natural form.

Extremum of the Response Function	Driving Speed, km/h	Distance to the Tree Crown, m	Working Fluid Consumption, L/h
Y_optimal_ = 661.99	*X*_1_ = 0 (3)	*X*_2_ = −0.441 (1.28)	*X*_3_ = −0.135 (37.3)
Y_optimal_ = 666.445	*X*_1_ = −0.591 (2.114)	*X*_2_ = 0 (1.5)	*X*_3_ = 0.11 (42.2)
Y_optimal_ = 677.769	*X*_1_ = −0.57 (2.145)	*X*_2_ = −0.408 (1.296)	*X*_3_ = 0 (40)

**Table 7 sensors-22-03164-t007:** Analysis of the device operation in the greenhouse.

Droplet Size, Microns	Number of Drops after 0.1 min	The Number of Drops after 2 min	The Number of Drops after 10 min	The Number of Drops after 20 min	The Number of Drops after 30 min
2	6202	1833	1744	1278	374
4	3548	93.1	56	5.6	0.9
6	595	29.5	30	3.2	0.3
8	164.2	4.8	2.7	0.3	0.1
10	66.6	1.8	0	0	0
12	19	0.6	0.3	0	0
14	11.8	0	0	0	0
16	1.2	0	0	0	0
18	0.6	0	0	0	0
20	0.6	0	0	0	0
22	0	0	0	0	0

## Data Availability

Not applicable.

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
