# Peer review of "Automated Mobile Hot Mist Generator: A Quest for Effectiveness in Fruit Horticulture"

_sensors, 2022, doi:10.3390/s22093164_

Round 1

Reviewer 1 Report

Abstract

"It was revealed that in a single pass of the automated mobile
platform along the aisle, the density of droplets is 480 pcs/cm2." I would omit this highly technical data in an abstract

Introduction

"fruits and fruits". Repeated word, correct.

"it is necessary to almost double its increase." Insert a literature reference.

"Usually, best results are based on the following indicators: retention of droplets on the leaves". It's necessary to refer to consistent literature, "usually" is not a good scientific reference.

Literature Review and Problem Statement

This section must be improved delivering to the reader more concepts. Currently it sounds like a list of other studies, but it's not clear what the state of the art is in general terms.

"The main difference between cold and hot types of mist is the size of the resulting particles. Small droplets easily penetrate the crown of trees, evenly distributing through-out the treated surface. However, due to heating followed by evaporation, this type of generator is not compatible with all preparations." All this part must be referenced to literature.

"This, however, leads to a decrease in spraying effi-ciency and an increase in the drift of droplets of working fluid. To reduce the drift and improve the quality of spraying, forced deposition of preparations is used, for which ro-tating nozzles are installed on rod sprayers, together with low-power fans." Also here, a reference to literature is needed.

Automated technologies for sprayers. This chapter has to be improved. It sounds not clear for the average reader, and it partially overlaps in contents with the following chapter.

Materials and methods

"Water-sensitive paper (cards) were attached to apple leaves at different heights of 1 m, 1.5 m, 2 m and at different depths with a step of 0.3 m in the crown of the tree shortly before spraying. " Is this method new? Is this method validated by other studies? Insert literature references.

"coded (columns 2-4) and natural (columns 5-7) form" . It's difficult to read the table, mayvbe insert column numbers

Results.

"the density of the droplet coating was determined when introducing PPP in various operating modes of the mist generator (Fig. 5)." How many cards have been used and measured? How many trees? How many per tree? Is the method of the cards a validated one?

"corresponds to the distribution norm according to agrotechnical requirements." You must insert a reference to the norm or regulation.

Figures 7 and 8 are not clear at all, especially 8. If appropriate, split in more graphs and give more explanation of the representation.

"The coefficients of the mathematical model are found. " What are you going to optimize? Only the density of the droplets? What about fuel consumption, total water, total drug used, portion of the drug dispersed into the environment? You don't have to optimize all the parameters, but you have to clearly state what you are missing and what you are focusing on.

"following modes are most effective for performing the operation of ap-plying plant protection products using a mist generator installed on a automated plat-form: the speed of the automated platform is 2.1 km/h, the distance to the crown ....." Again, what are you optimizing? Are you accounting for the environmental impact(energy used, water used, PPP drug dispersed, ...)?

Discussion

"Moreover, it was found that up to 8% of the losses of the working fluid is due to
droplets smaller than 10 microns that evaporated in the air, up to 12% of the losses are
due to droplets carried outside the processing zone by air currents, up to 14% are losses
on large droplets, more than 100 microns, that did not stay on the sheet due to gravity." How did you determine these data? It's not clear.

Author Response

Дякую за конструктивні коментарі! ми внесли максимальні корективи

Reviewer 2 Report

The article presents a proprietary automated mobile platform with an aerosol generator of hot mist. The subject is interesting and relevant to the field of this journal. The authors should make some improvements to the manuscript in order to be suitable for publication.

The language should be further improved by a native English speaker.

The abstract does not provide the reader with information about the results. It has only one numeric value of the results. It needs to be improved, giving more numeric values for the results.

The introduction provides all the necessary information to the reader. But on the other hand, the references are not new. The authors should cite new references (last 5 years).

Also, at the end of the introduction, the authors don’t provide any information to the reader about the meaning of this work. What is the gap that this work is coming to fill? Please add a couple of paragraphs about the scientific contribution of this work.

I am a little confused about the reference numbering. Is there a reason the authors start with citations [59]?

The authors should change all the sentences in the text which are written in the first plural (line 190, 205, 491, 505, and 509).

It would be quite useful for the quality of this work to give some economic data for this system or compare this system with some similar systems.  

The Conclusions are too general, should give more useful conclusions. Maybe the authors should include some numerical values. This section should be rewritten more analytically.

Author Response

Thanks for the constructive comments! we have made maximum adjustments

Round 2

Reviewer 1 Report

Good improvements done.